# Effect of Different Timing of Local Brain Radiotherapy on Survival of EGFR-Mutated NSCLC Patients with Limited Brain Metastases

**DOI:** 10.3390/brainsci13091280

**Published:** 2023-09-03

**Authors:** Yu Wang, Shenghong Wu, Jing Li, Xiaohua Liang, Xinli Zhou

**Affiliations:** 1Department of Oncology, Huashan Hospital, Fudan University, Shanghai 200040, Chinaliangxiaohua@csco.ac.cn (X.L.); 2Department of Medical Oncology, Fengxian District Central Hospital, Shanghai 201499, China

**Keywords:** EGFR-TKIs, lung adenocarcinoma, brain metastases, brain radiotherapy

## Abstract

(1) Background: Epidermal growth factor receptor (EGFR) tyrosine kinase inhibitors (TKIs) have been the first line therapy for EGFR-mutant lung adenocarcinoma (LAC) patients with brain metastases (BMs). However, the role and the optimal time of brain radiotherapy remains controversial. We aimed to investigate the role of upfront brain stereotactic radiotherapy (SRS) and the impact of deferral radiotherapy on patients’ clinical outcomes. (2) Methods: We retrospectively studied 53 EGFR-mutant LAC patients with limited synchronous BMs between 2014 and 2020 at our institute. The limited BMs was defined with one to four BM lesions, with a maximal size of ≤4 cm. Patients were categorized into two groups: upfront brain SRS (upfront RT) and upfront TKIs. The intracranial progression-free survival (iPFS), progression-free survival (PFS), and overall survival (OS) between groups were analyzed. (3) Results: The median iPFS (21.0 vs. 12.0 months, *p* = 0.002) and PFS (20.0 vs. 11.0 months, *p* = 0.004) of the upfront RT group was longer than that of the upfront TKI group. There were no significant differences in median OS (30.0 vs. 26.0 months, *p* = 0.552) between the two groups. The upfront RT group is less likely to suffer from intracranial progression of the original sites than that of upfront TKIs during the disease course (36.1% vs. 0.0%, *p* = 0.025). Multivariate analysis showed that the Karnofsky Performance Scale and the presence of synchronous meningeal metastases were associated with overall survival. (4) Conclusions: Compared with upfront TKI, the combination of upfront SRS with TKIs can improve the iPFS and PFS in EGFR-mutant LAC with synchronous BMs. The addition of upfront brain SRS was useful for the original intracranial metastatic lesions.

## 1. Introduction

Lung cancer is still the leading cause of cancer-related death around the world [1]. Between 20 and 40% of non-small cell lung cancer (NSCLC) patients will develop BMs during the disease, especially for patients with EGFR-mutated adenocarcinoma [2,3]. The median overall survival of NSCLC patients with BM was only 7.0 months and 3–6 months for patients with diagnosis-specific graded prognosis assessment (DS-GPA) score of 0–2.0 [4].

Traditionally, local therapy strategies including SRS and WBRT or surgery resection still remains the main treatment for the brain metastasis [5]. However, the benefit of intracranial disease control from brain radiation was always yielded to neurologic toxicity. With the development of targeted therapy, EGFR-TKIs has been the standard first-line treatment for advanced NSCLC with BMs based on some famous Phase III trials [6,7,8]. The CNS objective response rate (ORR) of TKIs for NSCLC patients with EGFR-mutant and BMs was reported up to 70–93%, the median progression-free survival (PFS) and overall survival (OS) were 6.6 months and 15.9 months [9]. Given the high CNS control rate of TKIs and the neurologic function deficit from brain radiotherapy, the clinical use of cranial radiation has been suspected. There have been no Phase III clinical trials comparing the timing and efficacy of local brain radiotherapy with TKIs in patients with EGFR-mutated NSCLC with BMs.

Previously, many retrospective studies have investigated the difference in clinical outcomes between EGFR-TKIs alone and combination cranial radiotherapy with TKIs in EGFR-mutant NSCLC with BMs. The conclusions are inconsistent. Pre-clinical studies showed the coeffect of RT and EGFR-TKIs [10,11], and patients with BMs harboring EGFR mutations may have higher clinical response rates to WBRT compared to those with wild-type tumors [12]. Many meta-analyses suggested that brain radiotherapy plus EGFR-TKIs produced superior response rates and significantly prolonged the OS of NSCLC patients with BMs [13,14]. There were also meta-analysis that showed that upfront RT including stereotactic radiosurgery (SRS) was associated with better OS and iPFS for EGFR-mutated NSCLC with brain metastases [15,16,17]. Some clinical trials draw a conclusion that upfront brain RT based on first-line EGFR-TKIs might improve intra-cranial PFS but not OS in EGFR-mutant lung adenocarcinoma patients with BMs [18,19,20]. It is important to note that the radiotherapy technique involved in these studies are predominantly WBRT, whereas SRS is now the primary treatment option for patients with limited brain metastases.

The synchronous brain metastases, defined as the BM lesions occurring within 2 months of the primary cancer diagnosis and the prognosis is reported to be poor compared with metachronous lesions, which were defined as brain metastases found 2 months or more after the detection of the primary cancer [21]. A previous study suggested that the presence of EGFR mutations significantly correlates with greater edema and mostly a higher seizure incidence of BMs from NSCLC [22]. However, in clinical practice, patients diagnosed with treatment-naïve EGFR-mutant NSCLC without severe BM symptoms tend to choose EGFR-TKIs as first-line therapy and defer brain RT. As a result, some patients may be deprived of the opportunity to receive RT after they develop cranial progression disease after EGFR-TKI therapy. The benefit to OS for upfront RT based on specific patient characteristics, EGFR-TKI regimens, and radiation techniques remains unclear. Exploring the optimal management of BMs is of great importance in the era of TKIs. A randomized study evaluating the clinical outcomes between upfront EGFR-TKIs with radiotherapy (RT) at progression versus upfront RT, especially SRS, followed by EGFR-TKI is urgent. In this study, we conducted a retrospective study to investigate whether the combination of upfront brain SRS plus TKIs is more effective than upfront TKI in EGFR-mutant NSCLC patients with limited BMs.

## 2. Methods and Materials

### 2.1. Patient Population

A total of 219 patients diagnosed with NSCLC and BMs were treated in our hospital between 2014 and 2020. Treatment outcomes had been retrospectively evaluated in 67 patients receiving TKI treatment for EGFR mutations and BMs. All patients were required to meet the following inclusion criteria: (1) pathologically confirmed NSCLC harboring an activating EGFR mutation; (2) accompanied with synchronous limited BMs (AJCC stage IV disease) confirmed through magnetic resonance imaging (MRI) at first diagnosis or within 2 months; (3) Eastern Cooperative Oncology Group (ECOG) performance status ≤ 3; and (4) older than 18 years old. The limited BMs were defined as 1 to 4 BM lesions, with a maximal size of ≤4 cm. The exclusion criteria were set as follows: (1) patients who developed BMs after taking EGFR-TKIs; (2) patients who did not receive EGFR-TKIs after stereotactic radiosurgery (SRS); and (3) patients without sufficient information in the medical record and patients who underwent surgical resection of brain lesions at the time of initial BMs were also excluded. Finally, a total of 53 patients with limited BMs were included and were categorized into two groups: upfront brain RT plus TKI (upfront RT) and upfront TKI. The patients in the upfront TKI group may receive brain salvage RT or not when the brain disease progressed.

Clinicopathological data, including sex, age, history of smoking, histology, Karnofsky Performance Scale (KPS), EGFR mutation type, EGFR-TKI treatment, level of CEA in peripheral blood at initial diagnosis, number of brain metastatic lesions, leptomeningeal metastases, symptoms of CNS, number of extracranial metastases at baseline, and disease-specific Graded Prognostic Assessment (ds-GPA), were retrieved from patients’ medical records. The protocol was approved by the institutional review board of Huashan Hospital affiliated with Fudan University.

### 2.2. Treatment

All patients received oral administration of EGFR-TKIs, the dosages coming as follows: icotinib,125 mg/day, three times a day; gefitinib, 250 mg/day; erlotinib, 150 mg/day; and osimertinib 80mg /day.

The dose used for SRS was based on the institutional modification of Radiation Therapy Oncology Group protocols 90–05 and 95–08 [23,24]. The SRS dose was 24 Gy, 18 Gy, and 16 Gy for brain metastatic lesions with diameters less than 1 cm, 2–3 cm, and 3–4 cm, respectively.

### 2.3. Follow Up

Clinical and neuroimaging follow ups were performed at 3-month intervals. Neuroimaging examination was administrated by contrast-enhanced MRI. The therapeutic effect was evaluated by brain MRI, chest CT, and upper abdominal CT. Tumor response was assessed by the Response Evaluation Criteria in Solid Tumors (RECIST) 1.1. All data were collected from the medical record or by telephone.

### 2.4. Statistical Analysis

The iPFS was defined as the time from the diagnosis of BM to the time of intracranial progression or death without documented progression, the last follow-up time for patients who did not progress or died was a censored value. The PFS was defined as the time from the diagnosis of BM to any disease progression in the body or death without documented progression, the last follow-up time for patients who did not progress or died was a censored value. The OS was defined as the time from the diagnosis of BM to the time of death or last follow up if they were still alive. Survival analysis was performed using Kaplan–Meier curves. The effects of potential variables on OS were assessed using univariate analysis. Multivariate testing was performed by Cox regression analysis. The difference between the category variables was assessed using chi-square test. Statistical analyses were performed using SPSS software version 26.0.

## 3. Results

### 3.1. Patients’ Characteristics

A total of 53 EGFR-mutant lung adenocarcinoma patients initially diagnosed with BM were enrolled in the present study. Patients’ characteristics were shown in Table 1. Among them, 14 (26%) received combination therapy of upfront brain SRS plus EGFR-TKIs, and the other 39 (74%) received upfront EGFR-TKIs with 14 patients receiving salvage brain RT and 25 receiving TKIs alone. The median age of the population was 65 years old; 27 patients were male and 26 were female; the proportion of patients with KPS ≥ 70 was 66%; and 14 (26%) patients were tobacco smokers and 39(74%) were nonsmokers. An EGFR exon 19 deletion mutation occurred in 21 patients and an exon 21 mutation occurred in 27 patients. A total of 34 (64%) patients were initially diagnosed with extracranial metastases. Data showed that 29 patients (55%) had asymptomatic BMs at the beginning of their treatment. The upfront brain RT group had a high percentage of CNS symptoms in their BMs (28% vs. 93%, *p* = 0.000). The CNS symptoms including headache, dizziness, or obnubilation; seizures are not uncommon when accompanied with a series of CNS symptoms. Multifocal neurologic deficits including vision loss, hearing loss, limb weakness, and walking instability were also common manifestations. A total of 47 (89%) patients received the 1st or 2nd generation of EGFR-TKIs and the remaining 6 patients received the 3rd generation of EGFR-TKIs. The final follow-up date of the study was 1 November 2021. At the last follow up, 87% (46/53) of the patients had intracranial disease progression, 94% (50/53) had extracranial disease progression, and 68% (36/53) died.

### 3.2. Survival Outcomes

The median intracranial progression-free survival of the upfront brain radiotherapy group was longer (median 21.0 vs. 12.0 months, *p* = 0.002) than that of the upfront TKI group (Figure 1 upper). The progression-free survival of the upfront RT group was also superior to the upfront TKI group, the median time was 20.0 vs. 11.0months, *p* = 0.004 (Figure 2 upper). The upfront RT group showed longer OS than that of the upfront TKI group but without statistically significant (median 30.0 vs. 26.0 months, *p* = 0.522) (Figure 3 upper).

Among the 39 patients in the upfront TKI group, 14 patients were administered salvage brain RT when they exhibited brain disease progression, and the rest of the 25 patients did not receive brain RT until the last follow up. The time of brain RT had a significant influence on intracranial progression-free survival with a median iPFS of 21.0 months, 14.0 months, and 11.0 months (upfront RT vs. salvage RT vs. TKI alone, *p* = 0.001) (Figure 1 below). Similarly, the different timing of brain RT also significantly impacted progression-free survival (*p* = 0.011), and the upfront group had a significantly longer PFS than the other two groups with a median time of 20.0 months, 11.0 months, and 10.0 months (upfront RT vs. salvage RT vs. TKI alone, *p* = 0.011, Figure 2 below). The time of brain RT had a positive influence on OS with a median time of 30.0 months, 26.0 months, and 23.0 months (upfront RT vs. salvage RT vs. TKI alone, *p* = 0.477) (Figure 3 below). Also, the manifestation of brain MRI image in patients with upfront SRS are shown in Figure 4.

Univariate analysis showed that the KPS score, number of extra-cranial metastases, synchronous of meningeal metastases, and intracranial treatment during the course were associated with the overall survival (Table 2). Multivariate analysis showed that KPS (*p* = 0.01) and synchronous meningeal metastases (*p* = 0.000) were independently associated with the survival of patients (Table 3).

### 3.3. Patterns of the First Site of Failure and Intracranial Progression

Among the 53 patients, 50 patients suffered from different first sites of failure until the date of the last follow up, 20 patients had intracranial progression, 13 patients had extra-cranial progression, 10 patients suffered from both intra-cranial and extra-cranial progression, and 7 patients died without an effective record of disease assessment. The upfront RT group has a lower intracranial recurrence rate than the upfront TKI group (48.6% vs. 15.4%, *p* = 0.060). The upfront TKI group is more likely to suffer from intracranial progression of original sites than that of upfront RT during the disease course (36.1% vs. 0.0%, *p* = 0.025). Data are shown in Table 4 and Table 5.

## 4. Discussion

This retrospective study analyzed a consecutive cohort of EGFR-mutant NSCLC patients with synchronous limited number of BMs. To the best of our knowledge, this study focused on the application of upfront SRS plus TKIs in patients with synchronous limited BMs, excluding patients involved with upfront RT by WBRT. In addition, the effect of different timing of brain RT on the survival was also investigated. Current study showed that the combination of upfront brain SRS and TKIs significantly prolonged the iPFS and PFS of EGFR-mutant LAC with limited BMs and the deferral of brain RT until the progression of intracranial disease was associate with inferior iPFS and OS.

Many previous studies also reported similar results. Liu et al. also suggested that early brain RT with EGFR-TKIs may improve intracranial disease control compared with TKIs alone in EGFR-mutant NSCLC with BMs, and the addition of brain RT to EGFR-TKIs as initial therapy did not appear to improve survival in unselected patients, but in patients with low DS-GPA scores 0–2 [25]. A retrospective study by Manguson et al. investigated the impact of deferring radiotherapy in patients with EGFR-mutant NSCLC with BMs and showed that the use of upfront EGFR-TKIs, and the deferral of SRS or WBRT, may result in an inferior OS and a higher intracranial progression rate [26]. Miyawaki et al. draw a similar conclusion that upfront RT followed by EGFR-TKIs is more effective in terms of OS and PFS than upfront EGFR-TKIs for the survival of untreated patients harboring EGFR mutations with one to four BMs [27]. Moreover, there were many meta-analysis studies showing the superior clinical outcome of the combination of upfront brain RT with TKIs. Wang et al. conducted a meta-analysis including seven studies with 1,086 patients and concluded that, compared to TKIs alone, upfront brain RT and TKIs showed better iPFS and OS, especially for patients with a limited number of brain metastases [15,16]. Soon et al. reported the results from 12 non-comparative observational studies (n = 363) and found upfront cranial radiotherapy improved four-month intracranial disease PFS than TKIs alone [17]. Two randomized Phase II trials of Osimertinib with or without SRS in EGFR-mutant NSCLC with BMs (NCT03497767 and NCT03769103) are about to begin and we look forward to their results.

In the study of FLAURA and AURA3, Osimertinib showed more effective CNS efficacy and a reduced risk of CNS progression versus standard EGFR-TKIs in patients with untreated EGFR-mutated NSCLC [28,29]. Thus, Osimertinib has been recommended as a first-line therapy for EGFR-mutated NSCLC with BMs. Even in the era of Osimertinib, the role of upfront brain RT was also proved to be important. For example, Zhu et.al indicated that upfront SRS was independently associated with prolonged OS and PFS in patients with oligo-BMs in the Osimertinib-treated era [30]. Most of the patients (89%) in our study received the first or second generation of TKIs, which further certified the benefit of iPFS and PFS from upfront brain RT. Our study showed that the addition of upfront SRS plus TKIs insert a positive influence on the overall survival but without a significant difference, this may be due partly to the small number of samples in this study. In addition, the patients in the upfront RT group possessed a higher percentage of CNS symptoms (93% vs. 28%, *p* = 0.000) of which the tumor may associate with a larger volume, greater edema, and locations prone to cause functional impairment and thus result in a lower KPS score and an inferior OS. The lower proportion of patients with EGFR 19del in the upfront TKI group (31% vs. 64%, *p* = 0.026) may dilute the overall survival time. In the subgroup analysis of patients with EGFR exon 19 deletion, the upfront SRS also improved the iPFS and PFS than that of deferral RT and TKIs alone. However, the survival benefit of addition upfront brain SRS was not shown in the subgroup analysis of patients with EGFR exon L858R. This is consistent with the previous study, which showed that the superior benefits of EGFR-TKIs plus brain RT might be influenced by factors such as BM-related symptoms and mutation type, compared with EGFR-TKIs alone in the management of EGFR-mutated NSCLC patients with BMs. As for patients with EGFR exon L858R, more effective treatment strategies need to be further investigated for this subgroup of patients, including a combination of anti-angiogenesis or else.

Although the initial treatment of EGFR-TKIs can reduce the risk of CNS progression in patients with EGFR-mutant advanced NSCLC patients [31]. However, previous studies showed that first- and second-generation EGFR-TKIs had limited ability to cross the blood–brain barrier (BBB) [32,33]. Though the third-generation TKIs possess a better ability for BBB penetration, TKI resistance is still inevitable and the CNS is frequently the initial failure site after the clinical benefit from EGFR-TKIs [9,30,34]. Other studies have also shown the CNS to be the site of first progression in 61–85% of patients [35,36]. Our current study demonstrated an intra-cranial failure rate of 70.3% and the extra-cranial rate of 43.2% in the upfront TKI group, which were consistent with the results of previous studies [26]. The intracranial progression at the original site or a newly diagnosis of lesions remains a great challenge, which provides a strong rationale for the use of upfront RT, especially the SRS. This was supported in our study, as the intracranial progression of the original sites was 36.1% in the upfront TKI group and 0.0% in the upfront RT group. Our study indicated that patients with limited BMs in the upfront TKI group were more likely to develop meningeal metastases and multiple BMs at the first cranial progression during the treatment of EGFR-TKIs and thus losing the opportunity of SRS in salvage RT. Among the 14 patients in the salvage RT group, 6 patients received WBRT after developing meningitis and multiple BMs, while the remaining 8 patients had SRS after the progression of the original BM lesions. As for the 25 patients who did not receive any kind of RT after developing cranial PD, 6 patients were disabled for suffering from leptomeningeal metastases, and the others were denied SRS or WBRT due to the deterioration of their physical condition. In addition, the application of SRS can provide effective local control and associated with a lower cognitive decline and QOL (quality of life) impairment compared to WBRT [5]. This further supports the upfront SRS for BM in EGFR-mutant NSCLC patients with a limited number and volume of lesions.

Univariate analysis in the current study showed that the KPS score, number of extra-cranial metastases, the synchrony of meningeal metastases, and intracranial treatment during the course were associated with the overall survival. Multivariate analysis showed that KPS and the synchronous of meningeal metastases were independent factors of overall survival. Several previous studies demonstrated that patient KPS scores at initial diagnosis were independent factors for OS [26,37]. For patients accompanied with synchronous meningeal metastases, the prognostic is even poor with a median OS of only 3.0 months [38]. It should be pointed out that the therapy of patients with leptomeningeal metastasis from NSCLC is more complicated and deserves further investigation. The different treatment strategies including a combination of radiotherapy, intrathecal chemotherapy, systemic chemotherapy including new agents, and their optimal schedule should be prospectively re-evaluated in clinical trials.

Our study has several limitations that should be acknowledged. Firstly, this study was retrospective in nature and limited by all shortcomings of the retrospective study. Secondly, the neurologic toxicity associated with SRS was not evaluated because it was difficult to obtain from a retrospective study, though SRS is thought to preserve neurocognition for longer compared with WBRT. Thirdly, the number of patients in this study was limited, which may explain the superior OS of the upfront SRS group compared to the TKIs alone but without statistically significant. In the future, randomized and prospective clinical trials are necessary to investigate the efficacy and safety of radiotherapy plus EGFR TKIs and to determine the optimal treatment strategy for NSCLC patients with EGFR mutations and BMs.

In conclusion, upfront cranial SRS combined with EGFR-TKIs may improve intracranial-free survival and progression-free survival outcomes compared with upfront TKIs for EGFR-mutated NSCLC patients with limited synchronous BMs. The addition of upfront brain SRS was useful for the original intracranial metastatic lesions. Larger randomized trials investigating these two treatment strategies are urgently needed to identify optimal treatments for these patients.

## Figures and Tables

**Figure 1 brainsci-13-01280-f001:**
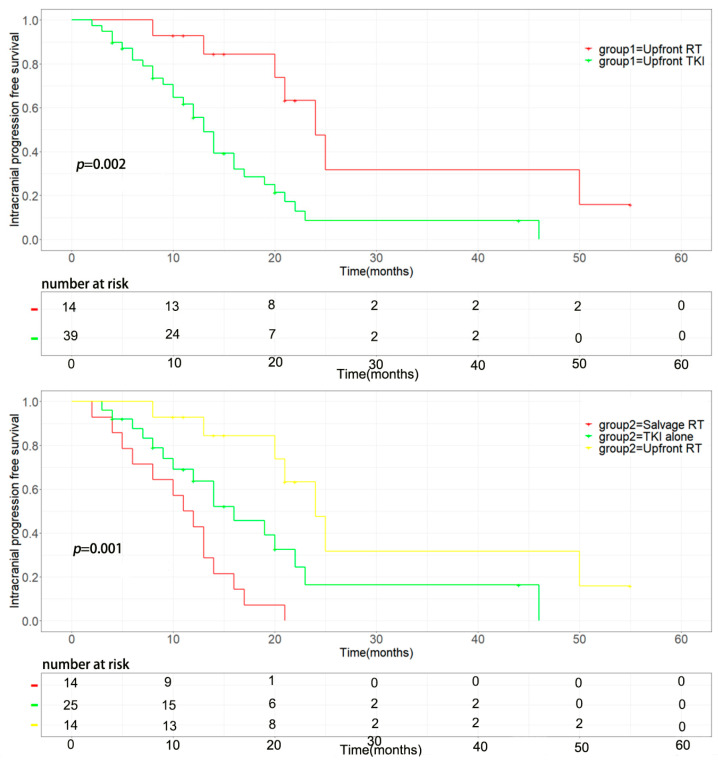
Intracranial progression-free survival of the groups. (**Upper**): the median iPFS of upfront RT group was longer than that of upfront TKI group. (**Below**): Impact of the timing of brain RT on intracranial progression-free survival (*p* = 0.001, *n* = 53): 14 patients were treated initially with brain radiotherapy (upfront RT), and 14 patients were administered salvage brain RT when they exhibited brain disease progression (salvage RT). Twenty-five patients did not receive radiotherapy until the last follow up (TKI alone).

**Figure 2 brainsci-13-01280-f002:**
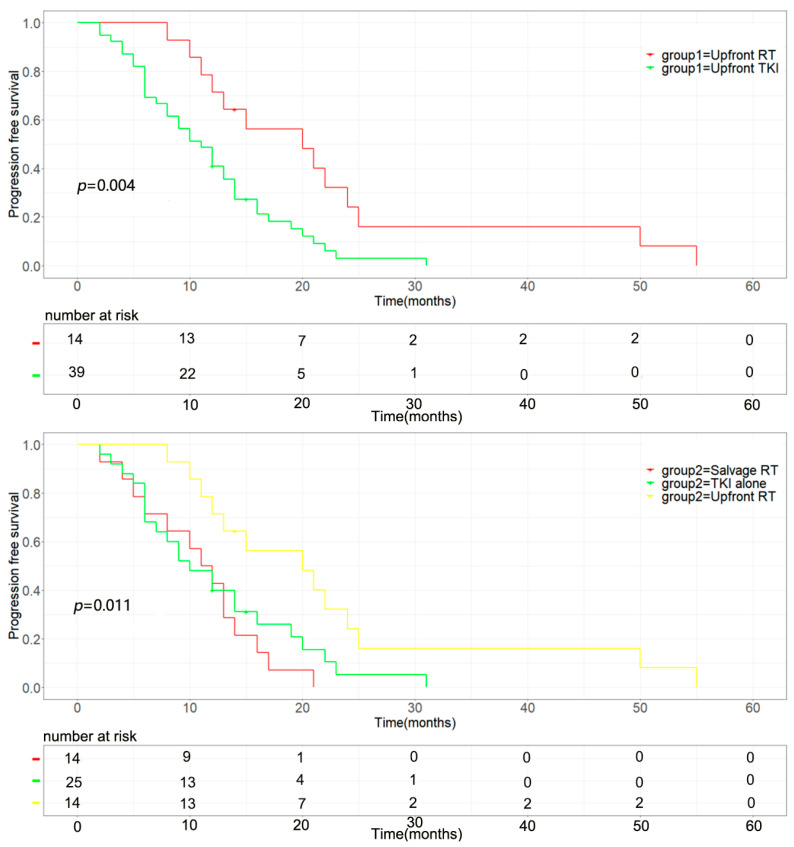
Progression-free survival of the groups. (**Upper**): the median PFS of upfront RT group was longer than that of upfront TKI group (*p* = 0.004). (**Below**): Impact of the timing of brain RT on progression-free survival (*p* = 0.011): 14 patients were treated initially with brain radio-therapy (upfront RT), and 14 patients were administered salvage brain RT when they exhibited brain disease progression (salvage RT). Twenty-five patients did not receive radiotherapy until the last follow up (TKI alone).

**Figure 3 brainsci-13-01280-f003:**
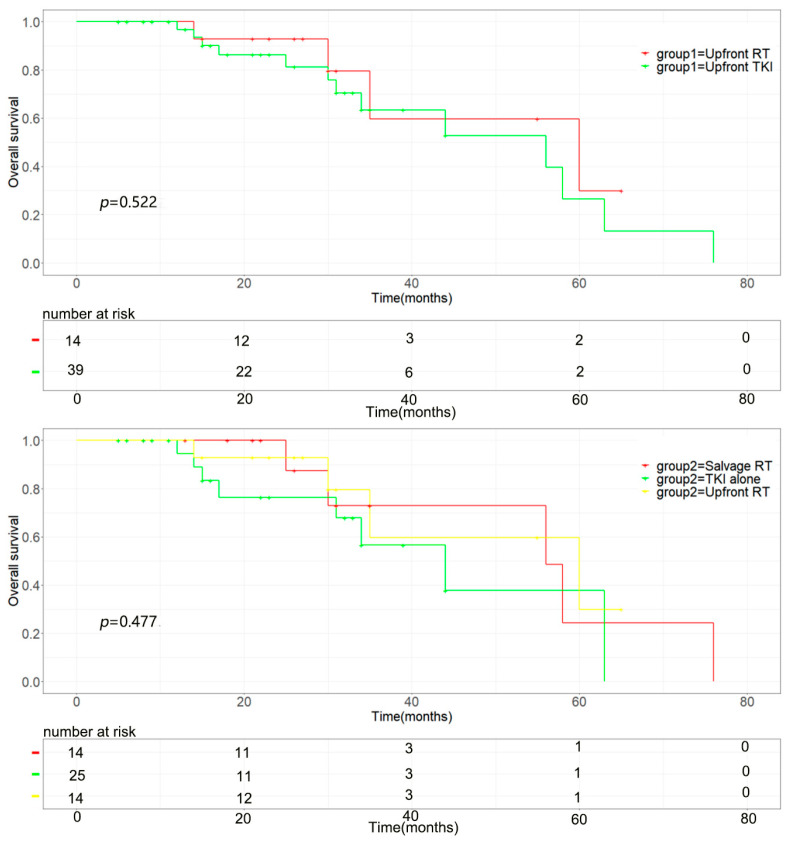
The overall survival of the groups. (**Upper**): The overall survival of upfront brain RT group was superior to upfront TKI group (*p* = 0.522). (**Below**): Impact of the timing of brain RT on progression-free survival (*p* = 0.477): 14 patients were treated initially with brain radio-therapy (upfront RT), and 14 patients were administered salvage brain RT when they exhibited brain disease progression (salvage RT). Twenty-five patients did not receive radiotherapy until the last follow up (TKI alone).

**Figure 4 brainsci-13-01280-f004:**
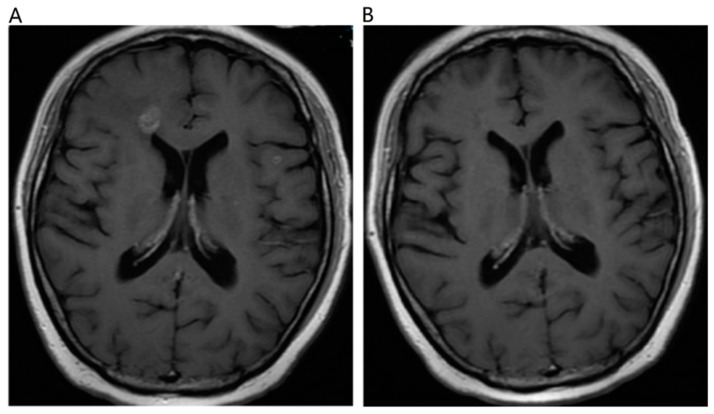
Manifestation on MRI image before and after patients’ brain metastasis lesions were treated with SRS. (**A**) The brain metastasis lesions before SRS; (**B**) the lesions were getting small after treatment of SRS.

**Table 1 brainsci-13-01280-t001:** Clinical and molecular characteristics of the patients.

Characteristic	Upfront TKI *n* = 39 (%)	Upfront RT *n* = 14(%)	*p* Value
Age, years			0.123
Median	65	58	
<60	13 (33.3)	8 (57.1)	
≥60	26 (66.7)	6 (42.9)	
Sex			0.253
Male	18 (46.2)	9 (64.3)	
Female	21 (53.8)	5 (35.7)	
KPS			0.230
<70	15 (38.5)	3 (21.4)	
≥70	24 (61.5)	11 (78.6)	
Smoking			0.630
Former/current	11 (28.2)	3 (21.4)	
Never	28 (71.8)	11 (78.6)	
EGFR mutation			0.026
19del	12 (30.8)	9 (64.3)	
L858R	22 (56.4)	5 (35.7)	
Other	5 (12.8)	0 (0)	
CEA of blood (ng/mL)			0.734
<12	16 (41)	5 (35.7)	
≥12	23 (59)	9 (64.3)	
Number of extra-cranial metastases			0.334
0	15 (38.5)	4 (28.6)	
1	12 (30.8)	4 (28.6)	
2	7 (17.9)	3 (21.4)	
3	4 (10.3)	2 (14.3)	
4	1 (2.6)	1 (7.1)	
CNS symptoms from BMs			0.000
With	11 (28.2)	13 (92.9)	
Without	28 (71.8)	1 (7.1)	
Generation of EGFR-TKI			0.012
1st/2nd	33 (84.6)	14	
3rd	6 (15.4)	0	
Graded Prognostic Assessment			0.760
0–2	32 (82.1)	12 (85.7)	
2.5–4	7 (17.9)	2 (14.3)	

EGFR, epidermal growth factor receptor; CNS, central nervous system; BMs, brain metastases; EGFR-TKI, epidermal growth factor receptor tyrosine kinase inhibitors; and CEA, carcinoembryonic antigen.

**Table 2 brainsci-13-01280-t002:** Univariate analysis for overall survival.

Characteristic	N	Median OS (Months)	95% CI	Log-Rank (*p* Value)
Age, years				0.767
<60	21	27.0	15.86–38.13	
≥60	32	27.0	16.64–37.37	
Sex				0.932
Male	27	27.0	14.92–39.08	
Female	26	30.0	22.17–37.83	
KPS				
<70	18	32.0	4.89–21.12	0.000
≥70	35	13.0	26.26–37.74	
Smoking				0.762
Former/current	14	27.0	11.94–42.06	
Never	39	27.0	14.02–34.21	
EGFR mutation				0.048
19del	21	39.0	7.89–70.11	
L858R	27	27.0	14.02–39.98	
CEA of blood (ng/mL)				0.053
<12	21	34.0	25.07–42.93	
≥12	32	23.0	16.25–29.75	
Number of extra-cranial metastases				0.007
0	19	39.0	24.63–53.37	
1	16	31.0	21.76–40.24	
2	10	23.0	20.11–25.89	
3	6	11.0	2.60–19.40	
4	2	11.0	-	
BM number				0.456
1	13	58.0	0.95–115.05	
≥2	40	56.0	30.99–74.44	
Time of meningeal metastasis				0.000
synchronous	7	15	7.97–22.03	
heterochronic	21	33	23.53–42.48	
CNS symptoms from BMs				0.717
With	24	30.0	24.31–35.69	
Without	29	26.0	16.33–35.67	
Intracranial treatment during the course				0.054
Yes	29	31.0	25.10–36.90	
No	24	22.0	12.33–31.67	
Generation of TKI				0.998
1st/2nd	47	27.0	20.02–33.98	
3rd	6	18.0	5.12–30.88	
GPA score				
0–2	44	30	20.87–31.13	0.018
2.5–4	9	49	36.76–61.24	
Upfront RT or not				0.552
Yes	14	30.0	25.81–34.19	
No	39	26.0	15.31–36.69	

KPS, Karnofsky performance scale; EGFR, epidermal growth factor receptor; CEA, carcinoembryonic antigen; CNS, central nervous system; BMs, brain metastases; RT, radiation therapy; SRS, stereotactic radiosurgery; OS, overall survival; CI, confidence interval; and GPA, Graded Prognostic Assessment.

**Table 3 brainsci-13-01280-t003:** Multivariate analysis of factors for overall survival.

Characteristic	β Value	SE Value	95% CI	Log-Rank (*p* Value)
KPS (<70, ≥70)	1.49	0.58	1.43–13.65	0.01
Meningeal metastasis (synchronous/heterochronic)	−2.49	0.71	0.02–0.34	0.000

KPS, Karnofsky performance scale; CI, confidence interval; SE, standard error.

**Table 4 brainsci-13-01280-t004:** Site of first progression of upfront RT and upfront TKIs (n = 50).

	Group		*p* Value
Site of first progression	Upfront TKIs (n = 37)	Upfront RT (n = 13)	0.741
Intra-cranial	18 (48.7%)	2(15.4%)	0.060
Extra-cranial	8 (21.6%)	5 (38.5%)	0.234
Concurrent	8 (21.6%)	2 (15.4%)	
Death	3 (8.1%)	4 (30.7%)	

RT, radiation therapy; TKIs, tyrosine kinase inhibitors.

**Table 5 brainsci-13-01280-t005:** Patterns of intracranial progression until last follow up (n = 46).

	Group		*p* Value
Site of intra-cranial progression	Upfront TKIs (n = 36)	Upfront RT (n = 10)	0.156
Original sites	13 (36.1%)	0 (0.0%)	0.025
Distant sites	11 (30.5%)	5 (50.0%)	0.253
Concurrent	6 (16.7%)	3 (30.0%)	
Death	6 (16.7%)	2 (20.0%)	

RT, radiation therapy; TKIs, tyrosine kinase inhibitors.

## Data Availability

The datasets generated and/or analyzed during the current study are available from the corresponding author on reasonable request.

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
