# Peer review of "Effect of Different Timing of Local Brain Radiotherapy on Survival of EGFR-Mutated NSCLC Patients with Limited Brain Metastases"

_brainsci, 2023, doi:10.3390/brainsci13091280_

Round 1

Reviewer 1 Report

In the article "Effect of different timing of local brain radiotherapy on survival of EGFR-mutated NSCLC patients with synchronous brain metastases" Wang et al. proposes a study comparing upfront tyrosine kinase inhibitors (TKI) therapy versus stereotactic radiosurgery (SRS) + TKI in EFGR positive long adenocarcinoma. Even if the study is retrospective and analyzes data related to 1st generation TKI and less data related to Osimertinib or other 3rd generation TKI, the study is well done and pleasant to read, and the results are courageous, even if sometimes they are not in accordance with the data reported in the literature. This very aspect opens up new themes for meditation about the subgroups of patients who will benefit from upfront SRS+TKI. I would still mention the difficulty of reading the graphics (very small writing and similar colors) and the lack of comments about the adverse effects of SRS and SRS+TKI (cognitive impairment, I think it deserves a brief mention).

Reviewer 2 Report

The authors present an interesting study regarding the treatment of NSCLC with brain metastases. Comparing RT with the TK-RT combination is a necessary investigation to understand the biological behavior of lung cancer brain metastases.

We suggest some small changes: material and methods should be written in lower case

we should better clarify how the diagnosis of brain metastases occurs by identifying synchronous ones from metachronous ones. It is also useful to report onset symptoms in relation to genetic and molecular expression. Was ALK also investigated? If you never come? EGFR-Driven Mutation in Non-Small-Cell Lung Cancer (NSCLC) Influences the Features and Outcome of Brain Metastases.J Clin Med. 2023 May 9;12(10):3372. doi: 10.3390/jcm12103372. PMID: 37240478; PMCID: PMC10219312.) I advise you to read this text and keep it in mind in the discussion. There is little discussion of the current literature in the discussion especially regarding the different opinions towards preventive RT in lung cancer. I suggest adding the literature review. It would be helpful to add an example MRI image.

Reviewer 3 Report

The authors tried to clarify the role of upfront SRS plus EGFR-TKI for EGFR-mutant NSCLC patients.

I had some comments for the manuscript.

1. The title "...oligo synchronous brain metastasis". What's the definition of oligo brain metastasis in the research.

2. The efficacy of osimertinib for CNS metastasis is not equal to 1st and 2nd generation EGFR-TKI. It's better to exclude patients with osimertinib as first line treatment.

3. The distribution of EGFR subtypes is not equal between two groups. There 64.3% of patients harboring Exon 19 deletion in upfront RT group, and only 30.8% in upfront TKI group. The condition would influence the clinical efficancy.

4. In figure 1, the authors categorized the brain metastasis to one number and >=2 numbers. What's the rationale.

5. In KM curves, the authors should label the numbers at risk.

6. In general, there are many published papers discussing this issue. What's the present manuscript's novelty and strengths.

Please check the manuscript by English language editing.
